# MACAW: A Causal Generative Model for Medical Imaging

**Vibujithan Vigneshwaran**                                    *vibujithan.vigneshwa@ucalgary.ca*
*University of Calgary, Canada*

**Erik Y. Ohara**                                                      *erik.ohara@ucalgary.ca*
*University of Calgary, Canada*

**Matthias Wilms**                                                    *wilms@umich.edu*
*University of Michigan, USA*

**Nils D. Forkert**                                                   *nils.forkert@ucalgary.ca*
*University of Calgary, Canada*

**Reviewed on OpenReview:** *https://openreview.net/forum?id=eYWO37oqQ4*

## Abstract

Although deep learning techniques show promising results for many neuroimaging tasks in research settings, they have not yet found widespread use in clinical scenarios. One of the reasons for this problem is that many machine learning models only identify correlations between the input images and the outputs of interest, which can lead to many practical problems, such as encoding of uninformative biases and reduced explainability. Thus, recent research is exploring if integrating *a priori* causal knowledge into deep learning models is a potential avenue to identify and overcome these problems. However, encoding causal reasoning and generating genuine counterfactuals necessitates computationally expensive invertible processes, thus restricting analyses to a small number of causal variables and rendering them infeasible for generating even 2D images. To overcome these limitations, this work introduces a new causal generative architecture named Masked Causal Flow (MACAW) for neuroimaging applications[1]. Within this context, three main contributions are described. First, a novel approach that integrates complex causal structures into normalizing flows is proposed. Second, counterfactual prediction is performed to identify the changes in effect variables associated with a cause variable. Finally, an explicit Bayesian inference for classification is derived and implemented, providing an inherent uncertainty estimation. The feasibility of the proposed method was first evaluated using synthetic data and then using MRI brain data from more than 23,000 participants of the UK biobank study. The evaluation results show that the proposed method can (1) accurately encode causal reasoning and generate counterfactuals highlighting the structural changes in the brain known to be associated with aging, (2) accurately predict a subject's age from a single 2D MRI slice, and (3) generate new samples assuming other values for subject-specific indicators, such as age, sex, and body mass index.

## 1    Introduction

Recent advances in medical imaging and the emerging availability of digital health records have resulted in an abundance of data in healthcare. This wealth of data, along with the continuing increase in computing resources, has helped the field of medical image analysis enter a new era — deep learning. This leap holds significant promise for disease prevention, diagnosis, and treatment planning (MacEachern & Forkert, 2021). However, the translation of deep learning techniques from academic research into clinical deployment has

---

[1]Code is available at `https://github.com/vibujithan/macaw-2D`.

been slow (Winder et al., 2024). The primary challenge often lies in the mismatch between the data used for training and the data encountered in real clinical scenarios (Castro et al., 2020). This mismatch often causes models that perform well in research settings to generalize poorly when applied in real-world clinical environments. Furthermore, the limited clinical translation of deep learning models can be partly attributed to their "black box" nature, which lacks inherent explainability for their decisions (Vercio et al., 2020).

Both of these issues can be primarily related to the way these models are trained. More precisely, discriminative deep learning techniques are designed to maximize accuracy on a given dataset. This approach encourages models to exploit all possible correlations in the data, including shortcuts, to improve performance on that specific dataset. As a result, these models often capture spurious correlations and uninformative biases present in imaging data (Souza et al., 2024). Thus, while they excel in research settings, they frequently fail to generalize effectively or provide meaningful explanations for their predictions when they are deployed in clinical settings.

One approach to better analyze and interpret the available medical data is to move beyond correlation-based analyses and instead utilize causally informed models. This approach has not only the potential to identify which variables (*e.g.,* age, sex, race, scanner type, *etc.*) causally affect medical images but also to what extent. Within this context, causal discovery methods address the question of *which* variables have an impact, while causal reasoning methods explore *how* they affect the images (Sanchez et al., 2022b). While causal discovery is beyond the scope of this work, we demonstrate in this work how true causal reasoning can be achieved using medical images through generative modeling. This causal generative modeling provides a powerful framework to explore variables related to the data-generating process in a transparent manner.

A straightforward approach to causal generative modeling is integrating a known structural causal model (SCM) into an existing generative framework. Pearl (2012) describes a causal model as comprising three levels of complexity, referred to as the "causal ladder". These levels, in order of increasing complexity, are association, intervention, and counterfactuals. *Association* pertains to correlations in the data and is solely focused on modeling the probability distribution in the dataset. *Intervention* relies on structural assumptions about the underlying data-generation process and involves exploring interactions with variables to observe how outcomes change on a population level. Many existing deep generative models, such as conditional VAEs, conditional GANs, and causal generative neural networks, only fulfill the requirements up to the intervention level. Lastly, *Counterfactuals* can be used to investigate hypothetical scenarios on an individual level, encompassing both interventional and associational inquiries. A counterfactual query essentially asks the trained causal generative model, "How would a subject's data appear if it had been acquired under different conditions?". The generation of counterfactuals not only provides causal insights into metadata and medical images but also holds significant potential in tasks with practical applications, such as fairness, bias mitigation, data augmentation, data harmonization, and digital twins (Pawlowski et al., 2020).

While the term "counterfactual" is frequently employed in literature, only a few models, such as Deep Structural Causal Models (DSCMs) (Pawlowski et al., 2020), Neural Causal Models (NCMs) (Xia et al., 2022), Hierarchical Variational Autoencoders (HVAE) (Ribeiro et al., 2023), VQ-VAE and generalized linear models (Peng et al., 2024), and Diffusion SCM (Diff-SCM) (Sanchez & Tsaftaris, 2022b), have the ability to produce causally-grounded counterfactuals by following the *Abduction-Action-Prediction* steps as defined by Pearl (2012). Other methods in this domain mostly focus on generating realistic adversarial images aimed at deceiving classifiers. This scarcity of true counterfactual models for images arises from the requirement for invertible deep networks to achieve genuine counterfactual generation. To date, only the normalizing flow model is invertible by nature and has been previously used for causal modeling (Pawlowski et al., 2020). In this series of studies, each causal variable within a graph is modeled by a separate conditional normalizing flow, necessitating multiple normalizing flow models to represent the complete causal framework.

Alternatively, our work demonstrates, for the first time, that a single normalizing flow model coupled with masked autoencoders is sufficient to efficiently model complex causal structures. While we only consider three causal variables in the experiments in this work, the approach presented can be easily extended to any number of causal variables by defining the corresponding adjacency matrix. Within this context, we demonstrate that combining a standard dimensionality reduction technique, such as kernel principal component analysis (KPCA), with normalizing flows can effectively encode a given causal structure. In theory, the KPCA

technique can be replaced with any dimensionality reduction technique. Within its medical context, a relevant previous study is the work by Wilms et al. (2022) on counterfactual image generation and Bayesian classification. The key difference to the work presented here lies in their exclusive focus on a setup assuming independence between conditioning variables, while the method presented in this work encodes a complex causal structure of interactions between variables using masked autoencoders. Thus, the main contributions of this work can be summarized as follows:

1. We present and evaluate a novel method named masked causal flow (MACAW) for encoding the causal structures of the data-generating process into a generative model.

2. Using this model, we generate new high-resolution brain images and evaluate counterfactuals associated with brain aging.

3. The model's explicit density estimation enables direct Bayesian classification, eliminating the need for a separate discriminative model.

## 2 Background

### 2.1 Causal graphical models

Causal graphical models employ nodes to depict variables and edges to define their connections within a directed acyclic graph (DAG) to provide an intuitive method for defining and exploring dependencies. The two important conditions, the Markov condition and faithfulness condition, ensure that conditional independence in the joint probability distribution is accurately reflected in the causal graphical model. Readers interested in these concepts can find in-depth information in the book by Peters et al. (2017).

Consider the $d$-dimensional random vector $\mathbf{x} = [x_1, \ldots, x_d]^T \in \mathbb{R}^d$, which follows the distribution $p_x(\mathbf{x})$. Let a causal graphical model $\mathcal{G}$ be a DAG containing $d$ nodes, each represented by $x_i \in [1, d]$; the connection between nodes is defined by adjacency matrix $A \in \{0, 1\}^{d \times d}$. It is essential to emphasize that the adjacency matrix of any topologically ordered DAG is always triangular. Assuming Markovianity, $\mathcal{G}$ is a valid representation of $p_x(\mathbf{x})$ if and only if the probability density $p_x(\mathbf{x})$ can be factorized as follows:

$$p_x(\mathbf{x}) = \prod_{i=1}^{d} p_x(x_i | \pi(x_i)) \tag{1}$$

Here, $\pi(x_i)$ denotes the set of parents of the node $x_i$, where $\pi(x_i) = \{x_j; A_{j,i} = 1\}$. The basic network structure (adjacency matrix $A$) is typically established by making use of existing causal knowledge in the field. In cases with no or limited existing causal knowledge about the data-generating process, causal discovery algorithms can be utilized for this purpose (Glymour et al., 2019).

The typical method for modeling a probabilistic causal model is to use Structural Equation Modeling (SEM) with random noise. The structural equation for each variable $x_j$ is defined as $S_j : x_j = f_j(\pi(x_j), n_j)$, where $n_j$ represents the mutually independent exogenous noise variable of the noise distribution $p_n$. In this formulation, the observational distribution of variables $p_x(\mathbf{x})$ can be conceptualized as being generated by sampling from a noise distribution $p_n$ and then applying a set of structural equations $S$ to the sampled values. This implies that the observed variables are influenced by both, the noise distribution and the causal relationships represented by the structural equations.

The causal calculus (Pearl, 2012) was created for the utilization of a causal model. Specifically, the do() operator enables an intervention in the model. When applying the do() operator, specific functions are replaced with a constant in an SEM. Similarly, in the corresponding DAG, the edges going into the target of intervention are removed, but the edges exiting the target are retained.

## 2.2 Normalizing flows

Normalizing flows are used to model complex probability distributions, denoted as $p_{\mathrm{x}}$, by applying a sequence of transformations $\mathbf{T} = \mathbf{T}_1 \circ \cdots \circ \mathbf{T}_k$ to a simple density prior $p_{\mathrm{z}}$. Transformations in $\mathbf{T}$ must be both invertible and differentiable to allow training of the model using the change of variables formula:

$$p_{\mathrm{x}}(\mathbf{x}) = p_{\mathrm{z}}(\mathbf{T}^{-1}(\mathbf{x})) \mid \det J_{\mathbf{T}^{-1}}(\mathbf{x}) \mid \tag{2}$$

Here, $\det J_{\mathbf{T}^{-1}}$ is the determinant of the Jacobian of the inverse transformations. Efficient model optimization hinges on the ease of computing this determinant. As a result, Jacobians are often designed as triangular matrices, allowing for computation in $\mathcal{O}(n)$ time. Various techniques have been introduced in the literature to achieve this triangular structure, with common approaches including the use of coupling and autoregressive functions.

Of particular relevance to this present work, we briefly describe autoregressive functions next. Autoregressive functions transform a variable $x_i$ using variables $x_1$ to $x_{i-1}$, which in turn constrains the Jacobian of the transformation to be lower triangular. This is similar to writing a multivariate density $p_{\mathrm{x}}$ as a product of univariate conditional densities:

$$p_{\mathrm{x}}(\mathbf{x}) = p_{\mathrm{x}}(x_1) \prod_{i=2}^{d} p_{\mathrm{x}}(x_i \mid x_{1:i-1}) \tag{3}$$

Following this step, the flow model is trained directly through maximum likelihood optimization using equation 2. More detailed descriptions about normalizing flows are, for example, provided by Kobyzev et al. (2021).

## 2.3 Related work

This subsection outlines the role of related works in shaping the development of the proposed MACAW model. Specifically, our research is built upon the foundations laid by Wehenkel & Louppe (2021) in their work on graphical flows and Khemakhem et al. (2021) in their study on causal flows. Theoretical underpinnings of our framework closely align with these studies, and our work builds directly on these established proofs.

Wehenkel & Louppe (2021) highlighted the similarity between equation 1 and equation 3, and argued that autoregressive transformations can be interpreted as a method to model a causal network with a predetermined node ordering. Conversely, in the case of a specific DAG with a predefined causal relationship, the autoregressive conditioners can be selectively masked to incorporate the causal relationship into the model. Based on this new perspective, they proposed the graphical normalizing flow technique, a new invertible transformation with either a prescribed or a learnable graphical structure to inject domain knowledge into normalizing flows. In a similar work, Khemakhem et al. (2021) suggested that SEMs and autoregressive flows are similar and introduced a framework called causal autoregressive flow (CAREFL) for causal discovery. Furthermore, they showed that normalizing flows can generate effective counterfactual queries due to their invertible nature.

Both of these studies primarily focused on identifying the causal structure or topology within a given dataset, operating with limited variables. Specifically, the counterfactual implementation only permitted coupling flows, accommodating two sets of independent variables. In contrast, the primary focus of our work is to develop a method for classification and counterfactual generation for higher-dimensional datasets, such as images, and to encode complex (non-autoregressive) causal structures into the flows. Consequently, a model is required that scales effectively for larger dimensions and allows the parallel execution of the flow. Therefore, we introduce a neural network called the causally masked autoencoder (C-MADE) in this work, which is inspired by the masked encoder developed by Germain et al. (2015). This implementation requires only a single forward pass to compute all causal dependencies and their respective conditional likelihoods, making it a computationally efficient density estimator compared to existing alternatives. Subsequently, multiple C-MADEs are arranged in sequence to form the Masked Causal Flow (MACAW), akin to how

Papamakarios et al. (2017) utilized stacked MADEs to create masked autoregressive flows. The following section provides a detailed explanation of this method.

## 3 Methods

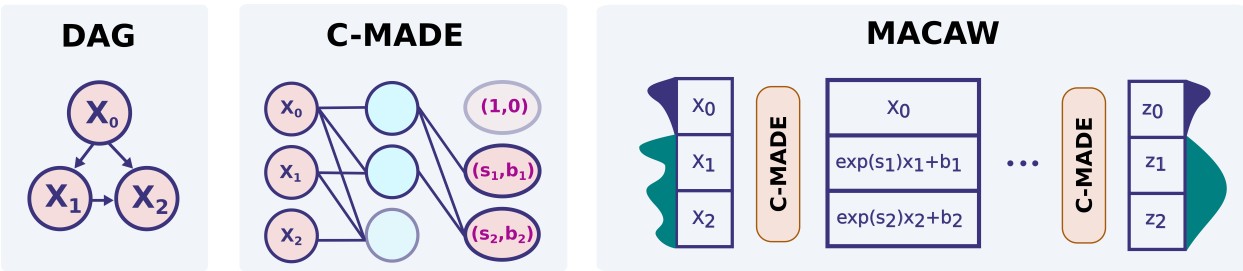

Figure 1: A causal DAG (left) and its respective C-MADE network (center). The MACAW architecture (right) consists of multiple C-MADE networks connected in a series, thereby forming a normalizing flow.

### 3.1 Causally-masked autoencoders (C-MADE)

We constructed C-MADE networks to efficiently represent the causal structure using a neural network for a given causal graphical model $\mathcal{G}$ with the adjacency matrix $A$. For simplicity, we assume that a neural network has the same number of input units ($\mathbf{x}$) and output units ($\mathbf{z}$). Since the output $z_i$ must depend only on its parents $\pi(z_i) \subset \mathbf{x}$, there must be no computational path between the output unit $z_i$ and any non-parent input units. A convenient way of zeroing connections is multiplying each matrix element-wise with a binary mask matrix derived from $A$, whose entries that are set to zero corresponding to the connections that need to be removed. Thus, the masks are essentially responsible for satisfying the causal structure.

Furthermore, we assume that our input is D-dimensional, *i.e.,* $\mathbf{x} = [x_1, \ldots, x_D]$, where the adjacency matrix $A$ characterizes the causal graph. To impose the causal property, we first assign an integer to each unit in the input and output layers from 1 to D. After numbering all the units, causal constraints on each unit are simply imposed by masking the connections based on matrix $A$.

While hidden layers can be added to this network, the causal parental structure has to be accurately maintained through the hidden layers. In this setup, $i^{th}$ hidden neuron in $j^{th}$ layer, $h_i^j$, is connected to its parents $\pi(h_i^j) \subset \mathbf{h}^{j-1}$ as well as $h_i^{j-1}$, which ensures that the learned hidden values are propagated to the final layer. This can be done by augmenting the adjacency matrix $A$'s diagonal units to 1 for all input-hidden layer connections and hidden-hidden layer connections. Moreover, the number of neurons in a hidden layer must be a multiple of $D$ to preserve the causal structure. If a hidden layer has $n \times D$ neurons, $A$ can be duplicated $n$ times and stacked to create the adjacency matrix for that layer. However, as the neurons in the final layer (output layer) are connected only to their parents, the network becomes strictly triangular, thus preserving the initial causal DAG structure. Figure 1 shows a causal graph and its corresponding C-MADE network.

### 3.2 Masked Causal Flow (MACAW)

Let us consider a normalizing flow model whose transformed probability density $p(\mathbf{z})$ is given as the product of prior distributions. The prior distributions for the source variables (variables that do not have parents) should be selected to resemble their original distributions closely. This is because the distributions of these variables are not modified throughout the training process. A standard normal prior is assigned to all variables that are not sources. Thus, a non-source variable $z_i$'s distribution is given by $p(z_i \mid \pi(z_i)) = \mathcal{N}(0, 1)$. This can be achieved by using the change of variable formula equation 2 with a series of affine transformations $T_t$, where $t \in [1, K]$. The transformed variables are denoted as $\mathbf{m}_t$. For notation consistency, we assign $\mathbf{x} = \mathbf{m}_0$ and $\mathbf{z} = \mathbf{m}_K$. Each affine transformation $T_t$ has scale and shift vectors, $\mathbf{s}_t$ and $\mathbf{b}_t$, estimated by a unique C-MADE network $C_t$.

$$(\mathbf{s}_t, \mathbf{b}_t) = C_t(\mathbf{m}_{t-1}) \tag{4}$$

$$\mathbf{m}_t = exp(\mathbf{s}_t)\mathbf{m}_{t-1} + \mathbf{b}_t \tag{5}$$

Since C-MADE is constructed as a causal network, the predicted affine transformation parameters of a variable can be considered as a function of its parents. Thus, the likelihood of the transformed probability density complies with equation 2. The negative log-likelihood of $p(\mathbf{z})$ was used as the loss function to optimize the C-MADE networks' weights. Following the optimization, the likelihood of a data point can be readily measured using a forward flow. Stacking multiple C-MADEs in a sequence provides the flexibility required to transform a complex distribution to the defined distribution (Papamakarios et al., 2017).

### 3.2.1 Generative sampling

To generate data using the trained model, we start by sampling the source variables from their prior distribution. Next, we proceed to sample each of the other variables iteratively by performing a backward flow. Moreover, in the case of interventional sampling, we have the option to set a particular value for a variable and continue sampling the remaining variables as usual.

### 3.2.2 Counterfactual inference

Counterfactual queries aim to assess statements about hypothetical situations of already existing observations. Let's assume that the MACAW was used to model the density of dataset X containing data vectors $[\mathbf{x}^0, \ldots, \mathbf{x}^N]$. Within this context, one observed data vector from the dataset is denoted as $\mathbf{x}^{obs} = [x_1, \ldots, x_D]$. For instance, if variable $x_j$ had taken the value $x_j = \alpha$ in our observed feature vector $\mathbf{x}^{obs}$, counterfactual queries can be used to determine what the value of variable $x_i$ would have been. This is denoted as $x_{i,x_j \leftarrow \alpha}$. According to Pearl (2012), generating causal counterfactuals requires three steps: *abduction*, *action*, and *prediction*. The *abduction* step evaluates the probability distribution over latent variables $\mathbf{z}^{obs}$ given observations $\mathbf{x}^{obs}$. In our model, this can be simply done using a forward flow, computing the transformation $\mathbf{z}^{obs} = \mathbf{T}(\mathbf{x}^{obs})$. The next step (*action*) is to intervene and fix the value of $x_j$ to a specific value $\alpha$, denoted as $do(x_j = \alpha \mid \mathbf{x} = \mathbf{x}^{obs})$, which makes it independent of its causes $\pi(x_j)$. In this step, the corresponding value of change of $x_{i,x_j \leftarrow \alpha}$ is adjusted in the transformed space $\mathbf{z}_j^{obs}$. The final step (*prediction*) is performed by computing an inverse transformation pass of the intervened $\mathbf{z}^{obs}$ to generate the counterfactual feature vector $\mathbf{x}^{cf}$, which is done through a backward flow. The algorithm 1 outlines the steps for the counterfactual process (Khemakhem et al., 2021).

---

**Algorithm 1** Counterfactual query

**Require:** observed data $\mathbf{x}^{obs}$, cf variable $x_j$, and cf value $\alpha$
  1. Abduction - forward flow: $\mathbf{z}^{obs} \leftarrow T(\mathbf{x}^{obs})$
  2. Action - change hypothetical values in the $\mathbf{z}$ space
  (a) $z_{j,x_j \leftarrow \alpha}^{obs} \leftarrow T_j(x_{\pi(j)}^{obs}, x_j \leftarrow \alpha)$
  (b) $z_{i,x_j \leftarrow \alpha}^{obs} \leftarrow z_i^{obs}$ for $i \neq j$
  3. Prediction - backward flow : $\mathbf{x}_{x_j \leftarrow \alpha} \leftarrow T^{-1}(\mathbf{z}_{x_j \leftarrow \alpha}^{obs})$

---

### 3.2.3 Bayesian classification

Let's denote our input distribution $p(\mathbf{x})$ as a joint distribution of a set of features $\mathbf{f}$ and a parent variable $c$ that we need to classify, which can take one value from $\{c^1, \ldots, c^R\}$. Thus, $p(\mathbf{x})$ can be written as $p(c, \mathbf{f})$. When we set $c = c^i$, the forward flow of the network provides the posterior of $p(c = c^i, \mathbf{f})$. This can be easily computed for all possible values of c $\{c^1, \ldots, c^R\}$ simply by setting the class variable accordingly. Thus, for a given class $c = c^i$, the posterior can be computed using Bayes' theorem as follows:

$$p(c = c^i \mid \mathbf{f}) = \frac{p(c = c^i, \mathbf{f})}{\sum_c p(c = c^j, \mathbf{f})} \tag{6}$$

Then, the class with the maximum a-posteriori (MAP) is chosen as the predicted class label. It is important to note that for each class variable, it is necessary to evaluate all possible posterior values and perform a forward pass with each of them.

### 3.3 Dimensionality reduction

As described earlier, normalizing flows are known for their resource-hungry nature. Consequently, to maintain the dimensions of the variables throughout the flows, a naive implementation of MACAW results in severe computational challenges. Therefore, when dealing with images, operating with all image pixels is impractical and even 2D images have to be projected onto a low-dimensional latent space before applying the MACAW model for density estimation. In this work, we utilized Kernel Principal Components Analysis (KPCA) to reduce the dimensionality of the training images. However, this technique can be easily replaced with any dimensionality reduction technique. The basic idea of this method is to project data that is not linearly separable onto a higher-dimensional space, thereby transforming it into a linearly separable form. We selected this technique for two primary reasons: (1) projected features are not linearly correlated with each other, as they are projections onto an orthogonal basis. This property reduces the dependence between projected image features, leading us to assume that this makes the optimization faster. (2) The inverse (preimage) of the projected KPCA features (here: polynomial kernel with a degree of 3) can be computed efficiently using existing algorithms.

### 3.4 Quantitative evaluation

Assessing the effectiveness of counterfactual image generation techniques poses challenges due to the absence of real-world ground truth. Monteiro et al. (2023) and then Melistas et al. (2024) proposed a framework for evaluating counterfactual images generated by various generative techniques. Their framework uses structural causal models and employs the counterfactual inference based on Pearl's *Abduction-Action-Prediction* steps to assess the performance of the different methods. In line with their study, we also conduct a specific set of experiments to evaluate the quality of our counterfactuals in terms of *Realism* and *Effectiveness*.

*Realism:* We used the Fréchet Inception Distance (FID) to quantify the semantic similarity between the generated counterfactual images and images in the training set. Therefore, real and generated samples were passed through an Inception v3 model (Szegedy et al., 2015) (pre-trained on Imagenet) to extract their semantic features and to calculate the FID between these two feature representations. A lower FID indicates that the features contain similar semantic information.

*Effectiveness:* Effectiveness aims to assess how well a counterfactual query performs. To quantitatively evaluate the effectiveness of a particular counterfactual image, we trained a separate discriminative deep learning model on the training set to predict the value of the intervened variable based on the image. While this traditional inference model may capture spurious correlations in the training data, it still provides information about the degree of confidence in the counterfactual generations.

## 4 Experiments and Results

We performed two experiments to investigate the effectiveness of MACAW. In the first experiment, we used synthetic data with a known causal structure to demonstrate that likelihood estimation, intervention, and counterfactual analysis function as expected. In the second experiment, we utilized the UK Biobank brain MRI images to showcase the practical applications and benefits of interventional sampling, counterfactual inference, and Bayesian classification that are only possible with a true causal deep learning framework.

### 4.1 Synthetic data

#### 4.1.1 Data

The first experiment aimed to evaluate if the proposed MACAW model can effectively learn the given causal structure within a dataset and accurately perform interventional and counterfactual queries. Therefore, we

created a synthetic dataset with predefined structural equations for sample generation to investigate this in detail. The causal structure of this dataset and the specific structural equations are defined as follows:

$$
\begin{aligned}
x_0 &= n_0 \\
x_1 &= n_1 \\
x_2 &= 2x_0 + x_1 + n_2 \\
x_3 &= 2x_0 + n_3 \\
x_4 &= 6x_2 x_3 + n_4
\end{aligned}
\tag{7}
$$

Where the noise variable $n_i$ was sampled from uniform $\mathcal{U}$ and normal $\mathcal{N}$ distributions as follows:

$$
\begin{aligned}
n_0 &\sim \mathcal{U}(0,1) & n_1 &\sim \mathcal{N}(1,1) & n_2 &\sim \mathcal{N}(0,2) \\
n_3 &\sim \mathcal{N}(0,0.5) & n_4 &\sim \mathcal{N}(0,0.1)
\end{aligned}
\tag{8}
$$

We generated 10,000 samples using equation 7 and partitioned it for training (70%) and testing (30%). Next, we defined prior distributions for the latent variables as follows: $z_0$ - drawn from a uniform distribution $\mathcal{U}(0,1)$, $z_1$ - drawn from a normal distribution $\mathcal{N}(1,1)$, while all other latent variables followed standard Gaussian distributions.

### 4.1.2 Training

For this experiment, our MACAW model consisted of 10 C-MADEs, each consisting of three hidden layers with 15 neurons each. During the training process, we utilized the negative log-likelihood as the loss function and halted training when reaching the point of minimal validation loss.

### 4.1.3 Generative sampling

After model training, we generated 10,000 random samples using the MACAW model. The distribution from SEMs and those generated using MACAW were very similar. Table 1 outlines the mean and the variance of the variables generated. We further compared the maximum mean discrepancy (MMD) between the ground-truth and generated datasets, obtaining a value of $3.187 \times 10^{-4}$, which indicates only a small distance between the two distributions. In addition, we have included the generated versus ground-truth data distributions in the appendix.

Table 1: Mean and the variance of generated samples using SEM and MACAW.

|  | $x_2$ | | $x_3$ | | $x_4$ | |
|---|---|---|---|---|---|---|
|  | SEM | MACAW | SEM | MACAW | SEM | MACAW |
| Mean | 4.01 | 3.99 | 2.99 | 2.99 | 73.96 | 74.55 |
| Variance | 5.32 | 5.92 | 0.58 | 0.53 | 2429.86 | 2863.50 |

### 4.1.4 Counterfactuals

To measure the accuracy of the counterfactual inference quantitatively, we estimated the counterfactual values for the action $\mathrm{do}(x_2 = 2 \mid \mathbf{x} = \mathbf{x}^{obs})$ in the test set. We computed the absolute difference between the observed values and the counterfactual values and then summed them up. Fig. 2(a) shows that the counterfactual query only affected $x_2$ and $x_4$ while other variables remained unaffected, as expected based on the causal structure depicted in equation 7. Next, we compared the counterfactual values with their respective ground truth value. These ground truth values were directly determined by substituting $x_2 = 2$ in the equation $x_4 = 6x_2 x_3^{obs} + n_4$, where the noise term $n_4$ was computed as $x_4^{obs} - 6x_2^{obs} x_3^{obs}$. Fig. 2(b) shows that the expected values of the ground truth and counterfactual values are perfectly aligned in the range of [-5,15], which is due to 99% of $x_2$ values falling within this range during training.

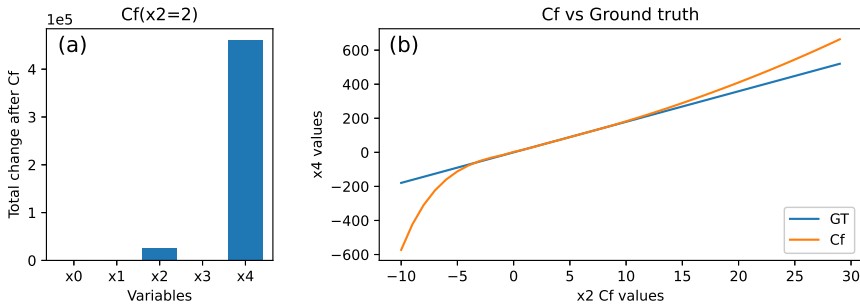

Figure 2: (a) The sum of the absolute difference between each variable in the test set after the counterfactual query for the action $\mathrm{do}(x_2 = 2 \mid \mathbf{x} = \mathbf{x}^{obs})$, (b) The expected values of the predicted counterfactual values and respective ground truth values.

### 4.1.5 Ablation Study: Causal vs. Correlation-Based Conditioning

A natural question is whether the causal masking mechanism in MACAW provides any advantage over a model that simply conditions on all available preceding variables. We address this question by training two MACAW models on the same synthetic data: one with the *true* causal DAG, and one with a *fully-connected* DAG in which every variable may depend on all variables that precede it in the topological ordering. The two models are identical in architecture, hyperparameters, and training procedure, and they differ only in the masks induced by their respective DAGs.

**Data-generating process.** We design a five-variable structural causal model with a strong confounding structure:

$$
\begin{aligned}
x_0 &\sim \mathcal{N}(0, 1), \\
x_1 &= 5\,x_0 + \epsilon_1, \quad \epsilon_1 \sim \mathcal{N}(0, 0.3), \\
x_2 &= 5\,x_0 + \epsilon_2, \quad \epsilon_2 \sim \mathcal{N}(0, 0.3), \\
x_3 &= 2\,x_1 + \epsilon_3, \quad \epsilon_3 \sim \mathcal{N}(0, 0.3), \\
x_4 &= 3\,x_2 + \epsilon_4, \quad \epsilon_4 \sim \mathcal{N}(0, 0.3).
\end{aligned}
\tag{9}
$$

The true causal edges are $\{0{\to}1,\ 0{\to}2,\ 1{\to}3,\ 2{\to}4\}$. Crucially, $x_1$ and $x_2$ share the common cause $x_0$ with a high correlation ($r \approx 0.988$). The fully-connected DAG adds all edges $\{i \to j : i < j\}$, introducing six spurious connections (most importantly $1 \to 2$), which allows the model to exploit the strong $x_1$–$x_2$ correlation as if it were a direct causal link.

**Observational fit.** Both models are trained on $N{=}10{,}000$ samples using identical hyperparameters (3 flow layers, hidden multipliers $[4, 8, 4]$, 800 epochs). As expected, both achieve comparable observational fit as measured by Maximum Mean Discrepancy (MMD) between generated and real samples (causal: 0.000723, fully-connected: 0.000652). The fully-connected model has strictly greater capacity (a superset of edges), so it can represent the observational distribution slightly better than the causal model. This confirms that observational metrics alone cannot distinguish the two approaches.

**Interventional predictions.** The key discriminating test is $\mathrm{do}(x_1 = v)$. Under the true causal structure, $x_2$ is not a descendant of $x_1$, and they are conditionally independent given $x_0$, so the intervention should leave $x_2$ unchanged at $\mathbb{E}[x_2] \approx 0$. We sweep $v \in [-15, 15]$ and plot $\mathbb{E}[x_2 \mid \mathrm{do}(x_1{=}v)]$ for both models (Fig. 3a). The causal model correctly produces a flat response, while the fully-connected model shows a strong linear trend as a direct consequence of the spurious $1 \to 2$ edge.

**Counterfactual predictions.** We further evaluate counterfactual queries of the form "What would $x_2, x_3, x_4$ have been if $x_1 = 10$?" Under the true causal structure, only $x_1$ and its descendant $x_3$ should

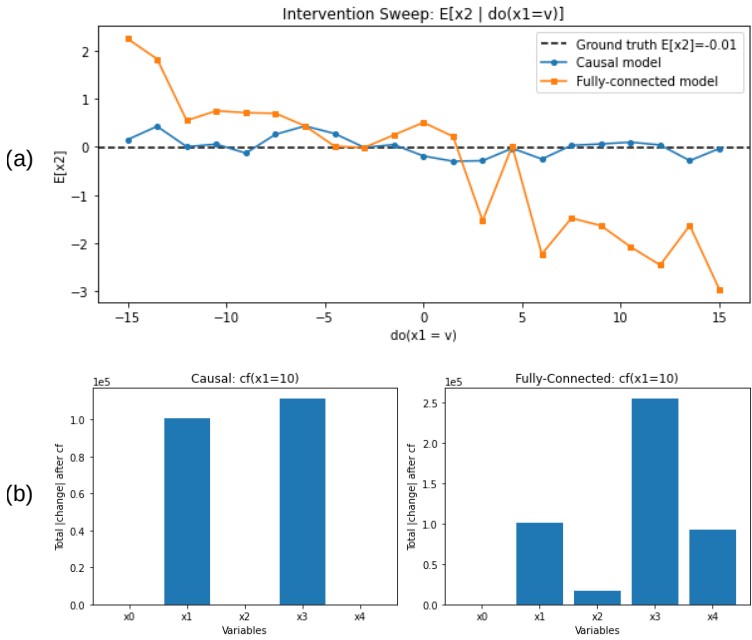

Figure 3: (a) When intervening on variable $x_1$, the causal model correctly yields a near flat response, whereas the fully connected model exhibits a strong linear trend, reflecting the spurious edge between the two variables. (b) When changing the variable $x_1$ counterfactually, the causal model correctly limits changes to $x_1$ and $x_3$, with negligible effects on the other variables. In contrast, the fully connected model introduces spurious changes in $x_2$, which further propagate to $x_4$.

change; $x_0$, $x_2$, and $x_4$ should remain at their observed values. Fig. 3b shows the total absolute change per variable after the counterfactual. The causal model correctly confines changes to $x_1$ and $x_3$, with near-zero changes in $x_0$, $x_2$, and $x_4$. The fully-connected model produces substantial spurious changes in $x_2$ (via the $1 \rightarrow 2$ edge) and consequently in $x_4$ (propagated through $2 \rightarrow 4$).

This ablation demonstrates that the causal masking in MACAW is not merely a structural prior that aids training, but it is essential for correct interventional and counterfactual inference. A model with more capacity (more edges) fits the observational distribution equally well but fails precisely where causal reasoning is required, because it cannot distinguish genuine causal effects from spurious correlations induced by confounders.

## 4.2 UK Biobank images

### 4.2.1 Data

The second and main experiment used neuroimaging data from the UK Biobank (UKBB) cohort (Miller et al., 2016). This study retrieved data under application 77508: Explainable and interpretable machine learning solutions in computational medicine. The UKBB's T1-weighted structural magnetic resonance imaging (MRI) used a 3D MPRAGE sequence with a 1-mm isotropic resolution and $208 \times 256 \times 256$ mm field of view. In the first step, starting with all subjects with available T1-weighted MRI data, participants with diagnosed brain-related disorders based on ICD10 codes (data field 41202, chapter V - Mental and behavioral disorders and chapter VI -Diseases of the nervous system) were excluded. We obtained the participants' sex information from the genetic sex data field (22001) and their age from the recorded values during the imaging visit (data field: 21003-2.0). Participants younger than 46 and older than 81 were excluded from the analysis because there was not a sufficient number of participants in these specific age groups to perform the age-stratified train-test split. Additionally, the body mass index (BMI) values were

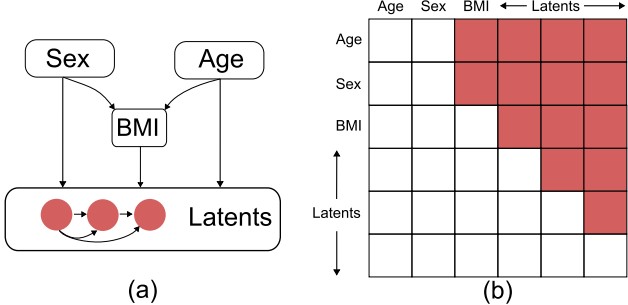

Figure 4: (a) Predefined causal graph for the UKBB dataset, incorporating age, sex, and BMI values. (b) Corresponding adjacency matrix, where filled cells indicate 1 and unfilled cells indicate 0.

retrieved directly using the UKBB data field 21001. Subjects with NaN values for either sex, age, or BMI were excluded from this work, resulting in a sample size of 23,692 (male = 11,050, female = 12,642).

All T1-weighted MRI data of the selected participants were aligned to the SRI24 atlas (Rohlfing et al., 2010), using the affine registration implemented in ANTs (Avants et al., 2011). From the 3D stack of data, we specifically chose the axial slice of a subject's image that primarily covered the lateral ventricular region, an area of the brain that captures the extent of visible atrophy due to aging. Finally, the data was split into training and testing sets of 80% and 20%, stratified by age.

### 4.2.2 Training

For model training, the extracted 2D images underwent center-cropping, leading to image sizes of $180 \times 180$ pixels. Subsequently, these images were projected onto a 1500-dimensional subspace (latents) using KPCA. Training all 1500 latents using the same MACAW network did not converge successfully, whereas training with 60 latents yielded a better likelihood estimation. Consequently, we divided the latents into subgroups of 60 and trained a separate MACAW model for each group, optimizing the likelihood individually for each model. Each MACAW model incorporated the causal structure illustrated in Figure 4. In this setup, age and sex were treated as discrete distributions, while BMI was considered continuous. The prior distributions for sex and age were determined based on the distribution of the training data, adopting Bernoulli and categorical distributions, respectively. Gaussian distributions were used as priors for both BMI and the latents. A 10% validation set was taken from the training set during the training phase to determine the best early stopping criteria.

### 4.2.3 Generative sampling

Generative sampling was conducted in an autoregressive manner using the trained models. The process began by randomly sampling age, sex, BMI, and 60 components from the first model. This process continued iteratively until 1,500 components were generated, which were then reconstructed by estimating the preimage of the KPCA components. During interventional sampling, specific values for one or more parent variables (age, sex, BMI) were manually defined, while the remaining variables were sampled from the first model. This autoregressive sampling approach was then repeated using the subsequent models. The results from the generative and interventional sampling are presented in Figure 5.

### 4.2.4 Counterfactuals

Counterfactual inference was performed on data points within the test set. Using this setup, we aimed to estimate how an image would appear if the person had different biological characteristics, including age, sex, and BMI. Counterfactuals were generated independently for each model and then reconstructed, following a process similar to the generative sampling. Figure 6 visually demonstrates that age has an expected impact on the ventricular and sulci regions. More precisely, increasing the age tends to enlarge the ventricular area while decreasing the age variable reduces the ventricular volume. Furthermore, changes in the sulci

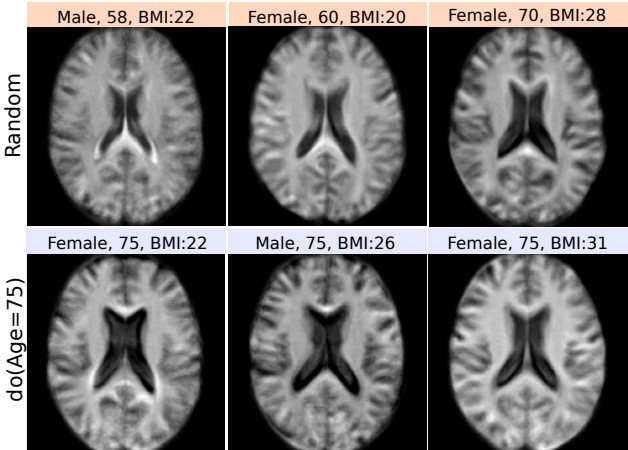

Figure 5: Results from two types of generative sampling: unconditional (top) and interventional (bottom). For interventional sampling, we set the age variable to 75 years.

regions are also observed when changing the age variable. When considering sex-related counterfactuals, variations in brain size and subtle changes (Lotze et al., 2019) in the ventricular regions can be observed in the figure. Furthermore, when generating counterfactual images to illustrate sex differences, the BMI values appropriately change, with males exhibiting higher BMI values. This is attributed to the causal influence of sex on BMI. Finally, BMI changes result in alterations of the lateral and ventricular parts of the brain, which may be related to accelerated brain aging (Beck et al., 2021). In addition, Fig.7 displays results from counterfactual queries simulating brain aging.

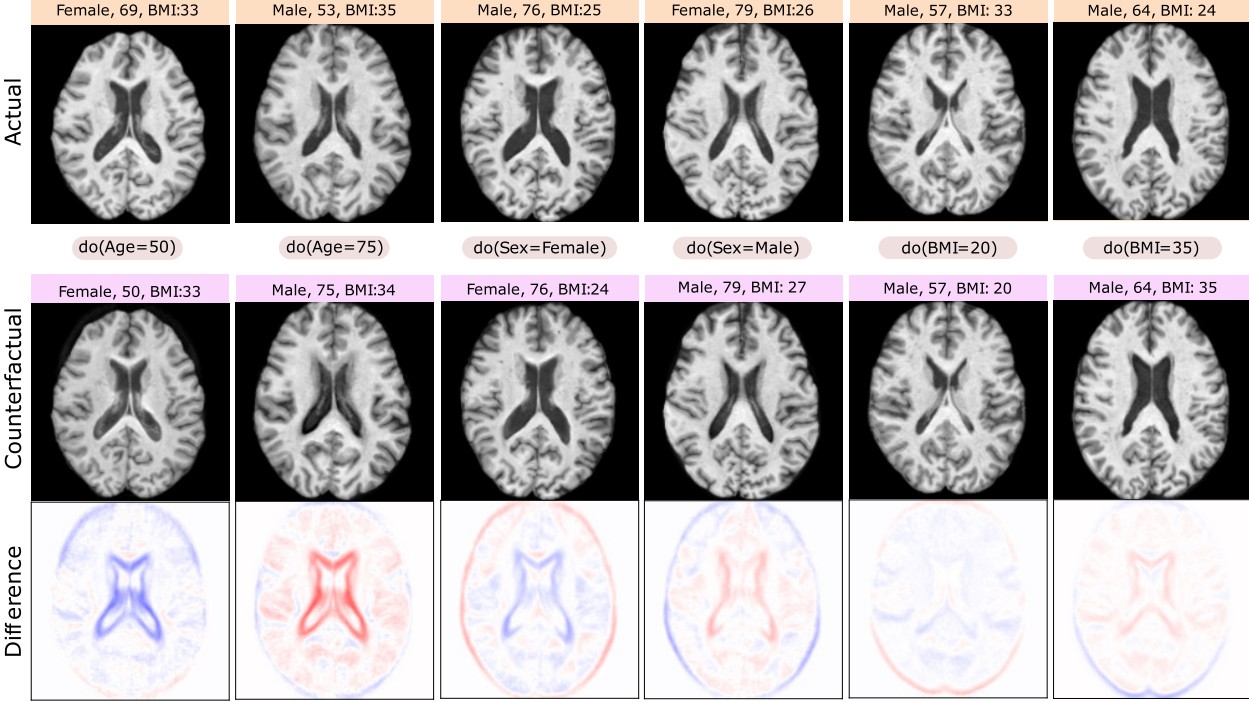

Figure 6: Illustration of the outcomes of counterfactual queries, where causal factors of randomly selected raw images were altered to generate new counterfactual images. Every column represents a distinct counterfactual query, identified by "do(.)" commands.

Table 2: Realism and effectiveness are measured on test, blurred, and counterfactual images. FID: Fréchet Inception Distance, MAE: Mean Absolute Error.

|  | Test | Blurred | $CF_{55}$ | $CF_{60}$ | $CF_{65}$ | $CF_{70}$ | $CF_{75}$ |
|---|---|---|---|---|---|---|---|
| Realism (FID) ↓ | 0.49 | 53.23 | 3.12 | 2.09 | 2.48 | 6.02 | 11.78 |
| Effectiveness (MAE) ↓ | 3.63 | - | 7.25 | 4.82 | 4.71 | 6.72 | 10.23 |

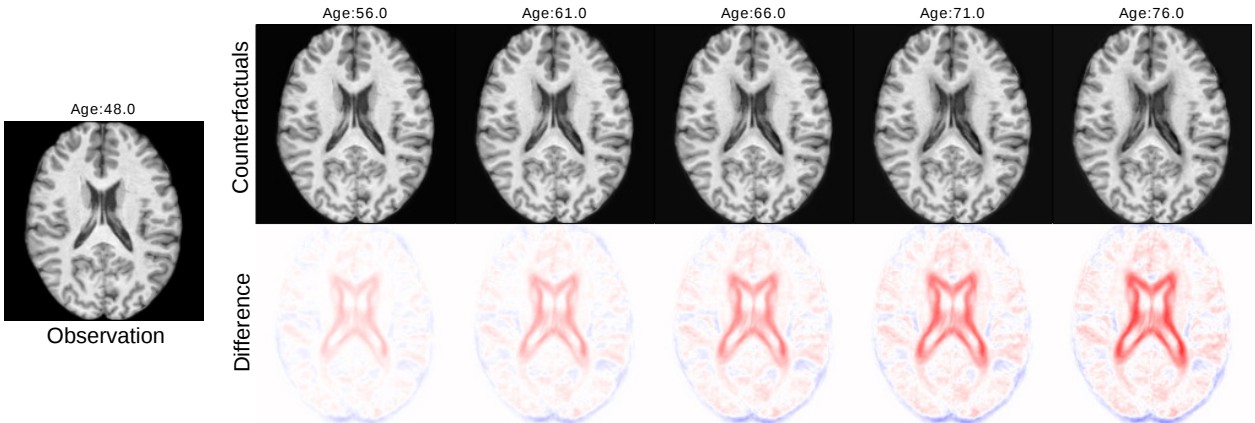

Figure 7: Results from counterfactual queries simulating brain aging. Each counterfactual image corresponds to a unique query "do($Age = x$)".

To assess the effectiveness of counterfactual inference, we generated counterfactual age values for the entire test set, consisting of 4,739 images. Essentially, the query asks what would happen if everyone in the test set had age $\alpha$, denoted as $CF_\alpha$, where $\alpha \in [55, 60, 65, 70, 75]$. The following experiments are conducted to assess these counterfactuals.

**Realism:** In the first step, the FID between the training and test sets was measured to serve as the lower bound. Subsequently, all images in the test set were blurred using Gaussian blur with a standard deviation of 1, and the FID between each original test image and smoothed test image was measured as a comparative baseline. Finally, the distance between each $CF_\alpha$ defined above and the training set was calculated. The results displayed in Table 2 demonstrate that the generated counterfactual images show superior *realism* compared to even slightly blurred images. Additionally, it was observed that *realism* was notably high for the counterfactual images generated for the age of 65, which aligns with the centered age distribution of the data.

**Effectiveness:** To assess whether the generated counterfactuals indeed contain predictive age information, we trained a separate classifier trained on the entire training set. Specifically, we utilized a popular architecture for brain age estimation, the SFCN model (Peng et al., 2021) trained for 2D slices, which achieved a mean absolute error (MAE) of 3.63 for test images. Subsequently, the $CF_\alpha$ sets were tested for age prediction using the counterfactual age $\alpha$ as the ground-truth target value for MAE computation. Table 2 displays these results, showing that $CF_{60}$ and $CF_{65}$ performed reasonably well. Since the UKBB has a large amount of training data for these age bins, it can be speculated that the model performs better when the queried counterfactual age is closer to the actual age (less severe changes on average). Therefore, to evaluate how the generated images change with counterfactual age differences, we computed the difference between the actual age and the counterfactual query age for all $CF_\alpha$ sets and subsequently measured the MAE value for this counterfactual age gap. Figure 8 illustrates these results, indicating that the counterfactual query performs well within the age range of -10 to +10.

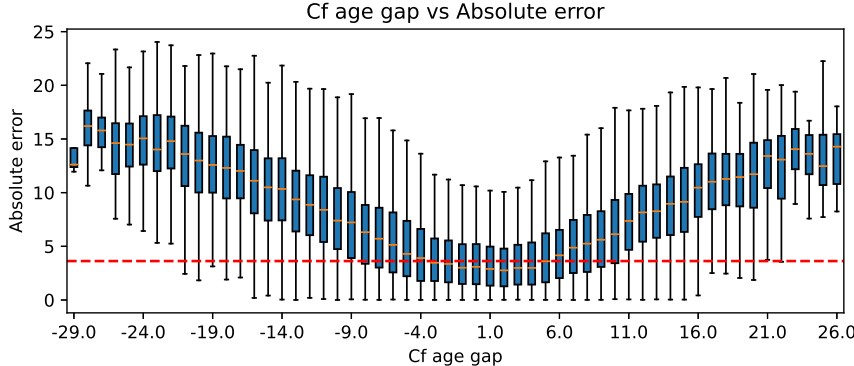

Figure 8: The figure displays how the effectiveness (MAE) of the generated images changes with the difference between actual age and queried counterfactual age. The dotted lines show the MAE of the test set (unaltered images).

### 4.2.5   Classification

Next, we employed the model's Bayesian classification abilities to predict the age of each participant in the test dataset as described in section 3.2.3. Therefore, we computed the posterior for each age and selected the age with the maximum a-posteriori (MAP) value as the predicted age. The overall accuracy of this prediction resulted in an MAE of 5.047 (standard deviation (std) = 0.052) when we used the first model (60 latents) for classification. Figure 9 illustrates the posterior distribution for a participant and the disparity between actual and predicted values. The figures show that (1) the prediction assigns higher posterior values around the true chronological age and lower for distant ages, and (2) the error in the prediction distribution is centered around 0. These two results indicate that the model identifies the causal connection between age and images quite effectively.

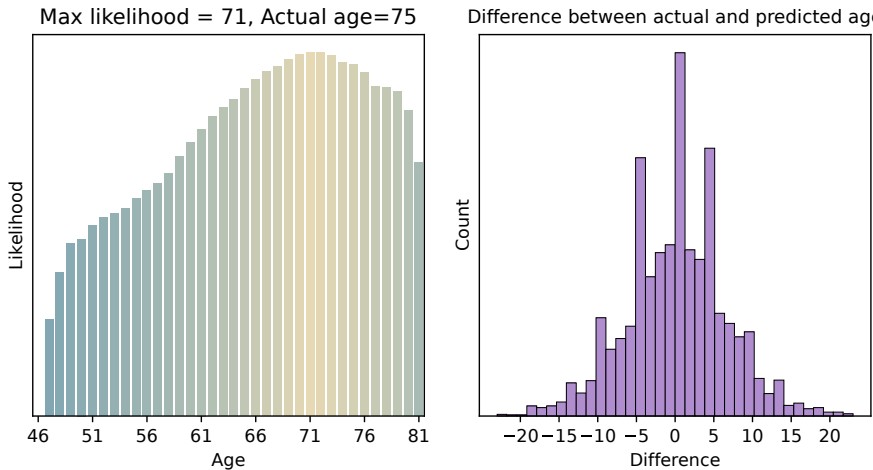

Figure 9: Posterior estimated by the model for each age, with the actual age being 75 and the predicted being 71 (left); distribution of the differences between the actual age and the predicted age for the whole test set (right).

## 5   Discussion

This work presents a novel causal generative framework called MACAW, which satisfies all three levels of the causal ladder (association, intervention, and counterfactual) and holds considerable promise for various

applications in medical and clinical contexts. Specifically, counterfactual generation offers causal insights at the individual level, which cannot be achieved using correlation-based approaches. Counterfactuals on the MRI data revealed brain regions associated with aging, sex differences, and BMI changes that are all in line with current knowledge.

Disentangling spurious correlations and identifying factors that causally influence medical images has been challenging. Discriminative deep learning models prioritize achieving accuracy on the training dataset, often exploiting shortcuts and spurious correlations. This makes it difficult to model and explore how variables like age, sex, and BMI causally affect medical images. In contrast, the causal generative modeling method described in this work encodes the data-generating process through a causal DAG, offering causal explanations at both, the population level (interventions) and the individual level (counterfactuals). Furthermore, causal counterfactual generation offers theoretical benefits with respect to fairness (Kusner et al., 2018), bias mitigation, data augmentation, data harmonization, and digital twin development. Furthermore, it is important to emphasize the distinction between conditional and causal approaches. In a standard conditional generation setup, the factors influencing image features are often assumed to be independent. For instance, when conditioning on the age variable, it is typically assumed that this has no impact on the BMI variable. However, in our setup, the model considers causal relationships within the dataset. As a result, when we intervene on the age variable, it correctly influences both, the BMI and the image features, reflecting genuine causal connections.

Quantitative assessments employing the metrics related to *Realism* and *Effectiveness* confirm that the generated counterfactual images effectively explain relevant causal information in the images. Furthermore, using interventional sampling, we created new samples with predefined parent values. This process can generate new samples that align with the characteristics learned from the population. Consequently, these generated samples can serve as supplementary data for training other deep learning models, particularly in cases where certain classes have insufficient data. However, potentially more relevant for knowledge discovery in the biomedical context, it is possible to use the method proposed in this work as the basis for digital twins.

While the primary purpose of our model is counterfactual generation, it can also perform classification using Bayes' theorem, similar to other density estimation methods. However, its classification accuracy is lower compared to CNN-based methods, like SFCN for brain age estimation (Peng et al., 2021). One potential reason for this finding is that, unlike task-specific models, such as the SFCN, our model's lower-dimensional subspace must capture features sufficient for accurate image reconstruction. However, the key advantage of our approach is that a single trained model can classify multiple variables (*e.g.,* age, sex, BMI), whereas CNN-based methods require separate models for each variable. Although classification is not the model's primary focus, it may serve as a solid foundation for developing discriminative deep learning models in the future.

Thus far, researchers have used VAEs (Pawlowski et al., 2020; Ribeiro et al., 2023), GANs (Nemirovsky et al., 2021), and diffusion models (Sanchez & Tsaftaris, 2022a; Sanchez et al., 2022a) for counterfactual image generation. However, for a causal model to truly perform counterfactuals, deterministic invertibility is essential (Pearl, 2012). Normalizing flows intrinsically offer this invertibility, and the experiments described in this work demonstrated that a single normalizing flow combined with masked autoencoders is effective in modeling complex causal structures. The proposed framework is grounded in solid mathematical foundations. In particular, using masking networks with a DAG-inducing adjacency matrix has been demonstrated as a valid causal generative prior in several previous studies, including Graphical Normalizing Flows (Wehenkel & Louppe, 2021) and Causal Autoregressive Flows – CAREFL (Khemakhem et al., 2021). The theoretical basis of our framework aligns closely with these prior works, and our approach builds directly upon their established proofs. While these previous studies primarily focus on causal structure discovery, our work extends these principles to generate image-based counterfactuals, representing a novel application in this domain.

Due to the high dimensionality of the image space, we projected images onto a latent space to enable a feasible model training. The dimensionality of the latent variables affects the model in two ways. First, it impacts the reconstruction of the final counterfactual image, as using fewer latent variables may lead to information loss. Second, it determines how effectively the causal parents can influence the generated images,

with larger latent spaces potentially diluting this effect. The ranked structure of principal components is particularly useful in this context. For instance, the first 50 principal components capture over 90% of the variance in the images in our experiments, while the remaining components were found to contribute only minor changes to the generated counterfactuals. Consequently, even if the causal influence of the parents diminishes the later latent components, it has minimal effect on counterfactual performance. An extensive ablation study on this topic has been conducted previously (Ohara et al., 2025).

This study has several limitations, with one of the most notable being the reliance on a predefined causal graph. In our research, we assumed the availability of a causal graph for the UKBB dataset. However, a predefined causal graph may not be readily available in many real-world scenarios. Typically, researchers need to estimate causal relationships from existing domain knowledge or through randomized controlled trials, which may not always be feasible or ethical. In cases where a predefined causal graph is absent, causal discovery techniques can be employed (Glymour et al., 2019). Within this context, it is important to emphasize that estimating causal relationships solely from observational and cross-sectional data can be an extremely challenging task, often even impossible. Additionally, to capture the complex structure of the brain associated with aging, it is necessary to extend our method to 3D images, which still remains as our future work. Another potential limitation of the proposed model relates to the computational demands of the classification task. Specifically, for each class variable, it is necessary to evaluate all possible values and perform a forward pass with each of them. Consequently, this process incurs a computational cost several times greater (equal to the number of classes) than the typical computational load associated with a standard discriminative network. Moreover, this classification process is limited to anti-causal prediction, meaning it involves predicting a top-level parent variable based on its effects.

In conclusion, the experimental results demonstrate the potential and effectiveness of MACAW in generating interventional and counterfactual images, as well as performing Bayesian classification. Future research should prioritize the development of a technique that seamlessly integrates dimensionality reduction and the normalizing flow framework into a single model. This integration, in turn, would facilitate the efficient processing of 3D images, ensuring that the model effectively captures all crucial information. The proposed technique holds potential for exploring and identifying potential new biomarkers for various diseases.

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

# A  Appendix

## A.1  Additional results for 1-D experiment

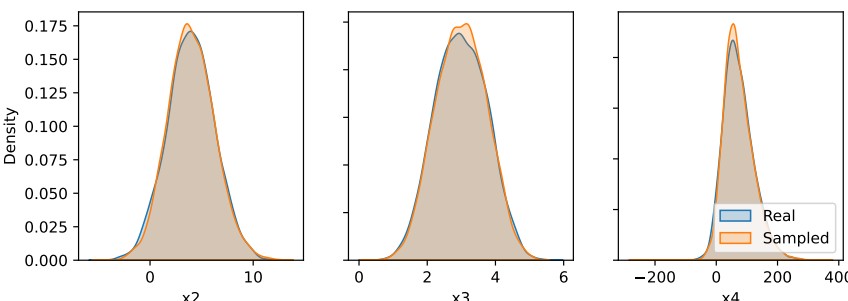

Figure 10: After model training, we generated 10,000 random samples using the MACAW model. As shown in the figure, the distribution from SEMs (ground truth) and those generated using MACAW were very similar.

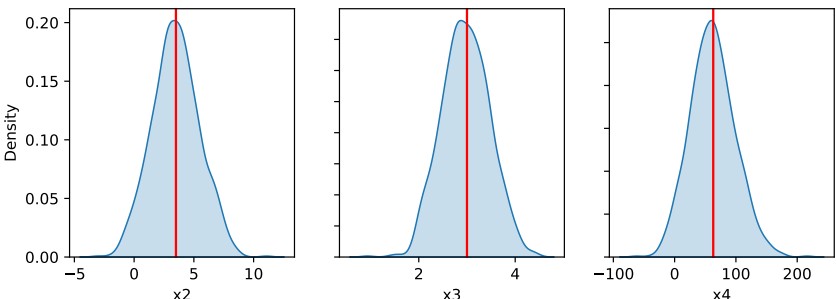

Figure 11: We generated samples from the model by intervening with two conditions: $\mathrm{do}(x_0 = 1.5 \mid \mathbf{x} = \mathbf{x}^{obs})$ and $\mathrm{do}(x_1 = 0.5 \mid \mathbf{x} = \mathbf{x}^{obs})$. The red lines represent the expected values as derived from the equations (the gold standard). It is evident that the distribution's mean closely matches these expected equation-based values.

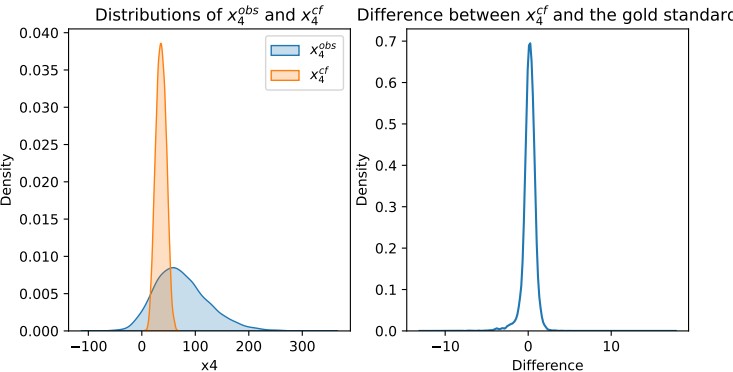

Figure 12: The figure displays both the observational and counterfactual distributions, along with the differences between the gold standard and the predicted values.

## A.2 MorphoMNIST benchmarking

For direct quantitative comparison with existing counterfactual image generation methods, we evaluated MACAW on the MorphoMNIST benchmark introduced by Melistas et al. (2024). This benchmark provides a standardized evaluation framework with established metrics, dataset splits, and corresponding baseline results for three model families: variational autoencoders (VAE), hierarchical VAEs (HVAE), and generative adversarial networks (GAN), all operating within a structural causal model (SCM) framework.

# B Models

VAE (Deep-SCM) (Pawlowski et al., 2020): A deep structural causal model that combines variational inference and normalizing flows to enable tractable inference of latent noise variables for counterfactual reasoning. It supports all three levels of causal analysis – association, intervention, and counterfactuals – and is validated on synthetic and medical imaging data.

HVAE (Ribeiro et al., 2023): A causal generative framework designed for high-dimensional data (*e.g.,* images), enabling accurate counterfactual and interventional inference. It introduces structured causal mechanisms and demonstrates strong performance in estimating direct, indirect, and total causal effects.

GAN (Taylor-Melanson et al., 2024): A causal generative approach for interpreting image classifiers through counterfactual explanations. It identifies influential features using Shapley-based methods and generates interpretable counterfactuals, showing improved explanation quality compared to existing tools.

### B.0.1 Data

MorphoMNIST (Castro et al., 2019) is a variant of MNIST where handwritten digit images are augmented with continuous morphological attributes. Following the benchmark protocol, we used the dataset provided by Melistas et *al.*, which contains 60,000 training and 10,000 test images of size $28 \times 28$ pixels with two continuous attributes: stroke *thickness* ($t \in [0.88, 6.26]$) and pixel *intensity* ($i \in [66.6, 254.9]$), along with the digit class label $y \in \{0, \ldots, 9\}$. Images were normalized to $[-1, 1]$, matching the benchmark convention.

### B.0.2 Training

The 20-dimensional encoded latents, together with the standardized thickness and intensity values and one-hot encoded digit labels, formed the input vector to the MACAW model. The causal edges for the MACAW model were defined to reflect the benchmark DAG: thickness $\rightarrow$ intensity, thickness $\rightarrow$ latents, intensity $\rightarrow$ latents, labels $\rightarrow$ latents, and autoregressive connections among the latent dimensions. The MACAW model was trained using $N_f = 4$ normalizing flow layers with hidden multipliers $[4, 6, 6, 4]$ for 200 epochs.

### B.0.3 Evaluation

We followed the benchmark's evaluation protocol with three metrics:

**Effectiveness.** For each test image, a counterfactual was generated by intervening on a single attribute ($\text{do}(t)$, $\text{do}(i)$, or $\text{do}(y)$) with a target value sampled from the training distribution, ensuring the target differed from the factual value. Three anti-causal CNN predictors (a thickness regressor, an intensity regressor, and a digit classifier) were independently trained on the real training images to predict each attribute from the images. These predictors were then applied to the generated counterfactual images, and effectiveness was measured as the mean absolute error (MAE) between the predicted and target attribute values for continuous attributes, and classification accuracy for the digit label as shown in Table 3.

**Composition.** The composition metric evaluates the stability of the encoding and then decoding under repeated null counterfactual operations (no intervention). For each test image, the cycle of latent space encoding, passing through the MACAW counterfactual mechanism with no attribute changes, decoding was repeated for $K$ cycles. The $\ell_1$ distance between the original and cycled images were measured at cycles $K \in \{1, 10\}$.

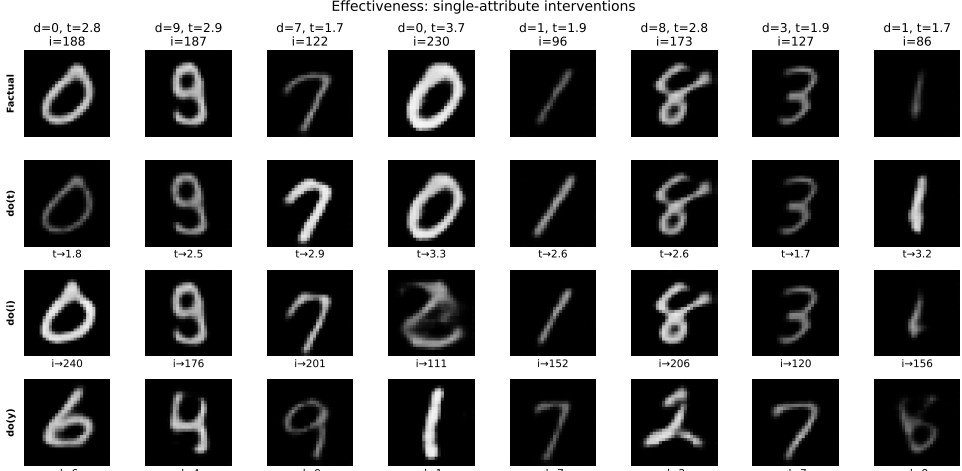

Figure 13: The figure shows the outcomes of counterfactual queries, where causal factors of selected test images (Factual) were altered to generate new images. Every row represents a distinct counterfactual query, identified as "do(.)" commands.

**Realism.** The Fréchet Inception Distance (FID) was computed between 10,000 counterfactual images (generated by intervening on thickness with random targets) and the real test images.

### B.0.4 Results

Table 3: Effectiveness on the MorphoMNIST test set. For each single-attribute intervention, all three attributes are measured: the directly targeted attribute and the preservation of non-intervened attributes. Thickness and intensity are reported as MAE ($\downarrow$), digit as classification accuracy ($\uparrow$). Values for HVAE, VAE, and GAN baselines are taken from Melistas et al. (2024).

|  |  | **MACAW** | HVAE | VAE | GAN |
|---|---|---|---|---|---|
| do($t$) | Thick. MAE $\downarrow$ | 0.179 | 0.086 | 0.109 | 0.228 |
| do($i$) | Intens. MAE $\downarrow$ | 15.39 | 3.52 | 5.33 | 15.14 |
| do($y$) | Digit Acc. $\uparrow$ | 0.93 | 0.972 | 0.775 | 0.451 |

Table 3 presents the effectiveness results. For the direct thickness intervention do($t$), MACAW achieved a thickness MAE of 0.179, compared to 0.086 for the H-VAE baseline. Under digit intervention do($y$), the model achieved a classification accuracy of 0.93 while preserving the morphological attributes.

Table 4: Composition, realism, and minimality on MorphoMNIST. Values for HVAE, VAE, and GAN baselines are taken from Melistas et al. (2024).

|  | **MACAW** | HVAE | VAE | GAN |
|---|---|---|---|---|
| Composition $\ell_1$ ($K=1$) $\downarrow$ | 0.0432 | 0.438 | 2.600 | 3.807 |
| Composition $\ell_1$ ($K=10$) $\downarrow$ | 0.1275 | 1.550 | 7.698 | 11.697 |
| FID $\downarrow$ | 16.66 | 5.362 | 10.124 | 35.568 |

Table 4 reports the remaining metrics. The composition scores indicate the stability of the encode–decode cycle, where lower values correspond to better image preservation under repeated cycling. Interestingly, the MACAW model outperformed the three baseline models, likely due to the invertible nature of the proposed network and the simple latent encoding.

It should be noted that the baseline methods (HVAE, VAE, GAN) in the benchmark employ deep generative architectures specifically designed for image generation, with each model containing a dedicated image

decoder that directly maps latent representations to pixel space. In contrast, MACAW operates on a compressed representation (d=20), with the normalizing flow modeling the causal relationships in this latent space. Despite this architectural difference, MACAW demonstrates competitive counterfactual generation quality, particularly in attribute preservation under single-variable interventions.

