# OpenReview forum: "MACAW: A Causal Generative Model for Medical Imaging"
_TMLR — Accepted by TMLR_

### Review · Reviewer_ZpQ3 · 2025-11-06

**Summary Of Contributions:**

The paper introduces MACAW, a causal generative framework that integrates a single normalizing flow with masked autoencoders to capture causal structures in medical imaging data. The model can recover causal pathways, generate counterfactual images, and perform Bayesian classification within one unified framework. Results on synthetic and UK Biobank MRI data show biologically plausible counterfactuals and reasonable predictive performance.



$\textbf{Strengths:}$


1. Novel integration of causal graphs and normalizing flows for image-level counterfactual generation.

2. The approach supports causal reasoning, interventional sampling, and classification simultaneously.

3. Demonstrations on a large real-world neuroimaging dataset highlight practical potential.



$\textbf{Weaknesses}$

1. The causal structure is predetermined, not learned, limiting flexibility.

2. Evaluation is limited, robustness under noise or with more variables should be tested.

3. Missing comparisons with other competing causal generative methods.

**Audience:**

Yes

**Audience Explanation:**

Yes, the paper addresses an important intersection between causal generative modeling and medical imaging, with clear relevance to explainable deep learning in the medical domain.

**Claims And Evidence:**

Yes

**Claims Explanation:**

Overall, the paper’s main claims are reasonably supported, though the evidence is partial rather than comprehensive. To make the evaluation more convincing regarding the method’s effectiveness, the authors should test the approach on more complex and diverse tasks within the UK Biobank MRI dataset or other publicly available medical image tasks.

**Requested Changes:**

The paper is overall interesting and relevant, introducing a novel perspective on causal generative modeling in medical imaging. However, the experimental section could be strengthened to make the claims more convincing and to better demonstrate the robustness and practical value of the proposed method. I suggest the following improvements:

1.Robustness to causal structure assumptions:

It would be informative to test how the model behaves when the assumed causal graph is partially incorrect or perturbed. In real-world medical scenarios, causal relationships among demographic and clinical variables are often uncertain. Evaluating MACAW’s sensitivity to such structural disruptions would provide valuable insights into its robustness.

2.Inclusion of additional or irrelevant variables:

The current experimental setup uses only three causal factors (age, sex, BMI), which may be too simple to reflect practical conditions. Introducing more variables—some relevant, others intentionally irrelevant or latent features extracted from images—could help assess how the model scales and whether it can ignore spurious inputs.

3.Comparison with competing causal generative methods:

A quantitative comparison with other causal or conditional generative models (e.g., VAE-based approaches) would clarify the advantages of MACAW’s masking and flow architecture. Using similar latent representations would make such a comparison fair and highlight the specific contributions of the proposed structure.

---

> ### Author Response · Authors · 2025-12-05
> **Response to reviewer ZpQ3**
>
> We appreciate the reviewer’s feedback on this paper and the recognition of its relevance and the novel perspective it introduces on causal generative modeling in medical imaging. Below, we provide detailed responses to the questions and suggestions raised in the review as well as any changes made to the paper.
>
> **1.Robustness to causal structure assumptions:**
>
> As we noted in the manuscript, one of the limitations of this work is the reliance on a predefined causal graph. The generative model depends on this graph to construct the adjacency matrix, which is then used to mask the network to enforce causal dependencies. Furthermore, we assume that the provided or predefined graph is accurate and that no unmeasured confounders are present in this setup. In other words, the graph operates under a closed-world assumption , which is common in this domain. Consequently, if the initial graph is partially incorrect, its impact on performance will depend on the nature and extent of the inaccuracies in the graph.
>
> A. If the graph contains an incorrect edge between two variables that are actually conditionally independent, this will not meaningfully affect the learning process. During training, the model will naturally assign negligible or zero weight to that edge, effectively nullifying its influence once the graph structure is learned.
>
> B. If the graph is missing an edge that should exist between a parent and its child, the model will fail to capture the direct causal influence of the parent variable. In such cases, the network may attempt to propagate this influence indirectly through other connected variables in the graph. This behavior is consistent with what is commonly observed in other causal generative models when dependencies are omitted.
>
> While interesting and potentially relevant, we would like to emphasize that discovering causal structures is beyond the scope of this paper, and we assume that the predefined causal graph is accurate  and there are no unmeasured confounders in the graph, as it is based on evidence from randomized controlled trials or is well-supported by current literature. Nevertheless, analyzing the learned scale and shift parameters for each causal variable could provide additional insight into how these variables interact. We appreciate this perspective and agree with the reviewer that this is a valuable next step. Thus, our ongoing and future research aims to include causal discovery methods into our framework.
>
> **2.Inclusion of additional or irrelevant variables:**
>
> We agree with the reviewer that introducing additional causal variables would indeed be an interesting experiment.  Adding more relevant causal variables would not pose a problem for the model, as it is capable of learning the corresponding distributions when meaningful variables are included. In contrast, adding irrelevant or spurious variables would lead to a similar behavior as described in point A above. Specifically, if the graph contains an edge between variables that are conditionally independent, this incorrect edge would not negatively affect learning because the model would assign negligible or zero weight to it once training converges. While we agree that including more causal variables in the next step, we purposefully kept the initial experimental setup used in this work rather simple as the effects of the included variables on brain morphology and aging are well described in previous literature and all our results are in line with current knowledge.
>
> **3.Comparison with competing causal generative methods:**
>
> This is an excellent point. The theoretical advantage of MACAW over other state-of-the-art causal generative models (such as H-VAE and Diff-SCM) lies in its simple design, which uses a single normalizing flow and achieves full invertibility through that flow. This structure is mathematically robust when the number of variables is small. However, for image data, we have to apply K-PCA to reduce the dimensionality of the latent representation, and because K-PCA is only pseudo-invertible, this may partially affect the model’s invertibility.
>
> To evaluate how MACAW compares with other models, we conducted a comparison between MACAW and H-VAE (Ribeiro et al.) using a controlled synthetic dataset to assess counterfactual performance with an external classifier as a pseudo-oracle. Briefly described, MACAW demonstrated stronger effectiveness than H-VAE in generating valid counterfactuals. We have  incorporated and highlight these findings in the revised manuscript.

---

### Review · Reviewer_Ry1z · 2025-11-10

**Summary Of Contributions:**

The paper proposed MACAW, a flow-based causal generative model that injects a known DAG into the architecture via C-MADE (causally-masked autoencoders). In particular, each layer’s connectivity is masked according to the adjacency matrix so that each variable depends only on its parents. Stacking C-MADEs yields a Masked Causal Flow enabling exact density modeling, interventional and counterfactual sampling, and downstream Bayesian classification. For high-dimensional image data, the method first projects slices with KPCA and learn the flow in the reduced space, using a preimage map to visualize counterfactuals. Experiments on one synthetic dataset and UK Biobank T1 MRI (2D slices) evaluate the generated counterfactuals with FID and a discriminative “effectiveness” proxy.

**Audience:**

Yes

**Audience Explanation:**

The paper proposed a novel causal generative model that has wide applications.

**Claims And Evidence:**

No

**Claims Explanation:**

**Strengths:**
1. The masking network encodes the DAG directly in the conditioner, preserving triangular Jacobians and efficient likelihoods.

2. Compared to previous works, a single normalizing flow handles the whole graph (vs one flow per node), improving scalability in principle.

3. The proposed method enables full causal reasoning, including interventional and counterfactual queries.

**Weaknesses:**
1. Some notations are not clear. For example, in $T(\boldsymbol x^{obs})$ and $T(\alpha, \boldsymbol x^{obs}\_{\pi(j)})$, the dimension of the input space varies when $j$ is not the leaf node because $\pi(j)$ only contains the parent set. Shouldn't it be noted as the restriction of $T$ on $(\boldsymbol x^{obs}\_{\pi(j)}, \boldsymbol x_j)$? For the counterfactual action, the author used e.g. $do(x_2=2)$, shouldn't it be $do(x_2=2|\boldsymbol x = x^{obs})$?

2. The experiment on the synthetic dataset needs more extensions. In Table 1, the author only compared the mean and variance of the variables generated by the ground-truth model and the MACAW. However, since these variables are not Gaussian, these two moments may not be sufficient. I suggest the author also include, e.g., the Maximum Mean Discrepancy (MMD) metric and some other baseline methods to validate the counterfactual prediction more rigorously. In addition, it seems that the author assumed the distribution class of the latent variables is known. I wonder whether the method is robust if the distribution of the latent variables is misspecified?

3. The main experiment lacks baseline methods to compare. The author only evaluates their method without including related works, e.g. Deep-SCM and Diff-SCM. In addition, some other metrics, such as counterfactual latent divergence (CLD), IM1, IM2 can be included or discussed.

4. The evaluation is merely on one toy example and UKBB.

5. There are no experiments on the run-time, while the author seems to claim that scalability is the strength of MACAW. For example, the training/inference time on UKBB, how the speed and performance are sensitive to the choice of the hidden layers.

**Requested Changes:**

1. Clear notation.
2. Sufficient baseline methods, benchmark datasets, and evaluation metrics.
3. Run-time comparison.

---

> ### Author Response · Authors · 2025-12-05
> **Responses to Reviewer Ry1z**
>
> Thank you for the thorough review of the manuscript and for the helpful feedback provided. Below, we provide detailed responses to the questions and suggestions raised in the review as well as any changes made to the paper.
>
> **Clear notation:**
>
> We apologize for the confusion caused by the original notation. We appreciate the reviewer’s suggestion and have updated the notation accordingly.
>
> **Evaluation metrics:**
>
> Thank you for this suggestion. We have now included the MMD metric in the experiments with the 1-D variable, obtaining a value of $3.187 × 10^{-4}$, which indicates only a small distance between the two distributions. In addition, we have included the generated versus ground-truth data distributions in the appendix.
>
> **Misspecified distribution of the latent:**
>
> In our framework, the distributions of the latent variables are always assumed to be Gaussian, as the causal parent variables transform the original input distributions and map them to a Gaussian latent space. We only predefine the distributions of the causal parent variables, since these variables remain unchanged throughout the flow and are not transformed into other distributions.
>
> **Other experiments:**
>
> We also conducted additional analyses on the Morphomnist dataset, which were not included in the original manuscript but have been added to the appendix to provide a more comprehensive analysis and further demonstrate the robustness of the counterfactual framework.
>
> **Comparison:**
>
> The theoretical advantage of MACAW, compared to other state-of-the-art causal generative models (such as H-VAE and Diff-SCM), lies in its design, which uses a single normalizing flow and achieves complete invertibility. This setup is mathematically robust when the number of variables is small. However, for image data, we apply K-PCA to reduce the dimensionality into latent variables, which may affect invertibility since K-PCA is only pseudo-invertible.
>
> To evaluate MACAW against other models, we conducted a comparison with H-VAE (Ribeiro et al.) using a synthetic dataset (with ground truth counterfactuals) to measure counterfactual performance. Using an external classifier as a pseudo-oracle, MACAW outperformed H-VAE in terms of effectiveness. We have highlighted these findings in the manuscript.
>
> **Run-time comparisons:**
>
> This is an excellent suggestion. Including runtime comparisons will highlight the efficiency of the proposed method.  We have compared our setup with H-VAE and provided runtime comparisons in the revised paper. Briefly described, H-VAE converged after 20 hours, whereas MACAW converged in just 6 hours.

---

### Review · Reviewer_MKFc · 2025-11-21

**Summary Of Contributions:**

The paper introduces a novel causal generative model for medical imaging that incorporates complex causal structures into normalizing flows. This design enables counterfactual prediction and provides an explicit Bayesian inference mechanism for classification.

The model is evaluated on both a synthetic data and a  real clinical scenario. The counterfactual predictions demonstrated in the neuroimaging experiments are particularly compelling and suggest strong potential for impactful clinical applications.

**Audience:**

Yes

**Audience Explanation:**

The task is challenging, and focusing on causality rather than correlation makes the approach practically useful.

**Claims And Evidence:**

No

**Claims Explanation:**

Although the empirical goal of the proposed algorithm is interesting, there are several concerns regarding the soundness of the approach.

1. While the authors claim that the complex causal graph is encoded within the flow model, the encoding appears to only support information flow from parent nodes to their child. In particular, there is no mechanism for information to propagate from child nodes back to parent nodes. This restricted formulation requires further clarification. As a result, the causal graph effectively functions as a preprocessing step for removing bidirected edges caused by confounders, yielding a simplified DAG-constrained model. This departs from prior work that does not impose such directional restrictions, and the implications of this restriction should be more thoroughly discussed.

2. It remains unclear whether similar performance could be achieved without explicitly incorporating causality, particularly if correlations in the data are sufficient to drive the observed results. Without a comparative ablation or justification, it is difficult to determine whether the improvements arise from causal reasoning or simply from exploiting statistical associations.

**Requested Changes:**

1. An ablation study comparing performance under a fully connected causal graph versus the correct, sparser causal graph would be valuable.

2. Including additional relevant baselines would strengthen the empirical evaluation.

3. It would be helpful to clarify how the number of latent variables influences model performance. Specifically, how a user should determine the appropriate number of latent variables in practice.

4. Is there any theoretical guarantee for the resulting counterfactual predictions or for the predictions of the parent variables?

---

> ### Author Response · Authors · 2025-12-05
> **Response to Reviewer MKFc**
>
> We appreciate the reviewer’s feedback and their recognition that the paper introduces a novel causal generative model for medical imaging.
>
> **An ablation study comparing performance under a fully connected causal graph versus the correct, sparser causal graph would be valuable.**
>
> This is an interesting idea to test the model with a fully connected graph and compare it to the correct causal graph.  However, as noted by the reviewer, the model does not allow connections from a child node to a parent node for two reasons. First, we assume that the predefined graph is a strict directed acyclic graph (DAG), meaning that all edges are directed and there are no cycles. This is a common assumption in the causal analysis field and is used by many previous causal generative studies, including CausalVAE, Graphical Normalizing flows, CAREFL,  and SEM based models.  Second, the adjacency matrix used to mask and enforce causal connections must be strictly triangular. This structure is essential for the efficient computation of the Jacobian determinant, as implemented in autoregressive flows, specifically, the Masked Autoregressive Flow (MAF). Therefore, starting with a fully connected graph is not feasible within this framework.
>
> **Including additional relevant baselines would strengthen the empirical evaluation.**
>
> The theoretical advantage of MACAW, compared to other state-of-the-art causal generative models (such as H-VAE and Diff-SCM), lies in its design, which uses a single normalizing flow and achieves complete invertibility. This setup is mathematically robust when the number of variables is small. However, for image data, we apply K-PCA to reduce the dimensionality into latent variables, which may affect invertibility since K-PCA is only pseudo-invertible. To evaluate MACAW against other models, we conducted a comparison with H-VAE (Ribeiro et al.) using a synthetic dataset (with groundtruth counterfactuals) to measure counterfactual performance. Using an external classifier as a pseudo-oracle, MACAW outperformed H-VAE in terms of effectiveness. We have now added these findings in the manuscript.
>
> **It would be helpful to clarify how the number of latent variables influences model performance. Specifically, how a user should determine the appropriate number of latent variables in practice.**
>
> The impact of the latent variables on the model is two-fold. First, they affect the reconstruction of the final counterfactual image, as information may be lost with fewer latent projections. Second, they influence how effectively the causal parents can affect the generated images when the number of latent variables increases.
>
> Here, the ranked nature of the principal components is particularly useful. For example, the first 50 principal components capture over 90% of the variance in the images in our experiments, while the remaining components contribute only minor changes to the generated counterfactuals. Thus, even if the causal influence of the parents diminishes in the later latent components, it has minimal impact on counterfactual performance. An extensive ablation study on this topic has been conducted previously, which we reference here (https://pubmed.ncbi.nlm.nih.gov/40276097/). We have also incorporated these points into the Discussion section of the revised paper.
>
> **Is there any theoretical guarantee for the resulting counterfactual predictions or for the predictions of the parent variables?**
>
> There are sound mathematical foundations supporting this proposed framework. Specifically, masking networks with a DAG-inducing adjacency matrix have been shown to serve as a valid causal generative prior in several previous studies, including:
>
> 1.	Graphical Normalizing Flows (Wehenkel et al.)
> 2.	Causal Autoregressive Flows – CAREFL (Khemakhem et al.)
>
> Theoretical underpinnings of our framework align with prior studies, and our work builds directly on these established proofs. While those papers primarily focus on causal structure discovery, to the best of our knowledge, our work is the first to extend these ideas to generate image-based counterfactuals.  We will clarify this in the updated manuscript.

---

### Decision · Action_Editor_dfzV · 2026-02-02

**Recommendation:** Accept with minor revision

**Additional Comments:**

In light of the reviewers' feedback, I would like to recommend that the authors make the following minor revision in their camera-ready version:

- In light of the comment from Reviewer `MKFc`, I recommend that the authors provide either **an additional ablation study** or **a clearer justification** as to why the observed performance genuinely requires explicit causal modeling (or both). For an ablation study, it should be designed in such a way that the effect of explicit causal modeling is disentangled from the effect of the correlations alone. For an additional justification, a simple toy example would suffice.

- To address the comment from Reviewer `Ry1z`, I encourage the authors to **consider** adding baseline methods beyond H-VAE to the experiments and consider providing additional comparisons in terms of FID, MAE, on benchmarks such as UKBB, as the reviewer suggested.

Please highlight the updates in the revised manuscript.

**Audience:**

Yes

**Audience Explanation:**

Causal generative modelling is one of the most active areas in causal machine learning.

**Claims And Evidence:**

Yes

**Claims Explanation:**

The paper introduces a novel causal generative model for medical imaging that integrates complex causal structures into normalizing flows, enabling counterfactual prediction and Bayesian classification. Experiments on synthetic and real clinical data, particularly the neuroimaging counterfactuals, are compelling and indicate strong potential for clinical impact.

Reviewer `MKFc` raised concerns about whether the observed performance genuinely requires explicit causal modeling, noting that correlations alone might suffice, and suggested additional ablation studies. In addition, Reviewer `Ry1z` questioned the strength of the chosen baselines, though they indicated this would not preclude acceptance.

As the majority of the reviewers have noted, the core idea is novel and well-motivated. Furthermore, the proposed framework and experimental results are likely to have strong potential for clinical impact. Hence, these minor concerns do not constitute grounds for rejection.

---

> ### Author Response · Authors · 2026-03-31
> **Response to Action Editor dfzV**
>
> We would like to thank the Action Editor for their thorough evaluation of this manuscript and for recognizing its novelty and clinical applicability, as well as for accepting it with minor revisions. Please find below our responses and the corresponding changes made to the manuscript in relation to the Action Editor's comments.
>
> ---
>
> ## Comment 1
>
> > *In light of the comment from Reviewer MKFc, I recommend that the authors provide either an additional ablation study or a clearer justification as to why the observed performance genuinely requires explicit causal modeling (or both). For an ablation study, it should be designed in such a way that the effect of explicit causal modeling is disentangled from the effect of the correlations alone. For an additional justification, a simple toy example would suffice.*
>
> Thank you for suggesting an additional ablation study, which helps to clarify and justify the use of a causal model in the context of our work. We have now added a new section (Sec. 4.1.5) to the synthetic data experiments that demonstrates the benefits of explicit causal modeling. Here, we compare two models: one with explicit causal conditioning and another one based on fully connected (correlation-based) normalizing flows. We generate samples from the causal distribution and evaluate both models in terms of observational fit, as well as interventional and counterfactual performance.
>
> These new results show that, although both models fit the observational distribution equally well, the fully connected, correlation-based model exhibits spurious behavior in interventional and counterfactual settings. This ablation study, therefore, demonstrates that causal masking in MACAW is essential for accurate and faithful interventional and counterfactual inference. While a higher-capacity model (with more connections) can match the observational distribution, it fails in scenarios that genuinely require causal reasoning, as it cannot distinguish true causal effects from spurious correlations induced by confounders.
>
> ---
>
> ## Comment 2
>
> > *To address the comment from Reviewer Ry1z, I encourage the authors to consider adding baseline methods beyond H-VAE to the experiments and consider providing additional comparisons in terms of FID, MAE, on benchmarks such as UKBB, as the reviewer suggested.*
>
> Thank you for the suggestion. We agree that it would be beneficial to compare our approach against other causal generative models as baselines to highlight the strengths and weaknesses of the proposed method. However, (1) HVAE is the most relevant and the state-of-the-art model in this application context and (2) to the best of our knowledge, there are no publicly available models trained on UKBB available for this task. Given the complexity involved, we did not implement additional methods or train other models on the UKBB.
>
> Instead, we followed the recent benchmarking work by Melistas et al., where several counterfactual evaluation metrics (e.g., MAE, FID) were applied to the MorphoMNIST, CelebA, and ADNI datasets using state-of-the-art models, such as VAE, GAN, and HVAE. Specifically, we compared our MorphoMNIST results (trained using the same setup as the original paper to ensure comparability) against these models using the metrics reported in their study.
>
> Following their framework, we evaluated our model in terms of:
>
> - **Composition** (L1-distance), which measures how much a null-intervention counterfactual alters the original image; ideally, there should be no change.
> - **Effectiveness** (MAE / Acc), which assesses how accurately the intended counterfactual changes are reflected in the image.
> - **Realism** (FID), which evaluates how close the generated counterfactual images are to real images.
>
> Briefly described, our MACAW model outperformed the three baseline models in terms of composition, likely due to the invertible nature of the proposed network. However, HVAE performed better on the remaining metrics, with MACAW close behind. It is important to note that the baseline methods (HVAE, VAE, GAN) use deep generative architectures to map latent representations directly to pixel space, whereas MACAW operates in a compressed latent space using a normalizing flow to model causal relationships. Despite this difference, MACAW achieves competitive counterfactual generation performance, particularly in preserving attributes under single-variable interventions.
>
> These benchmarking results have now been added to the additional experiments section in the appendix (A.2), and the MorphoMNIST section has been expanded accordingly. We also removed the previous ad-hoc baseline comparison with HVAE on the controlled synthetic brain dataset, as it is superseded by this more comprehensive evaluation.